# Assessing the Correlations between Different Traits in Copper-Sensitive and Copper-Resistant Varieties of Jute (*Corchorus capsularis* L.)

**DOI:** 10.3390/plants8120545

**Published:** 2019-11-26

**Authors:** Muhammad Hamzah Saleem, Shafaqat Ali, Mahmoud F. Seleiman, Muhammad Rizwan, Muzammal Rehman, Nudrat Aisha Akram, Lijun Liu, Majed Alotaibi, Ibrahim Al-Ashkar, Muhammad Mubushar

**Affiliations:** 1MOA Key Laboratory of Crop Ecophysiology and Farming System in the Middle Reaches of the Yangtze River, College of Plant Science and Technology, Huazhong Agricultural University, Wuhan 430070, China; saleemhamza312@webmail.hzau.edu.cn; 2Department of Environmental Sciences and Engineering, Government College University, Allama Iqbal Road, Faisalabad 38000, Pakistan; mrazi1532@yahoo.com; 3Plant Production Department, College of Food and Agriculture Sciences, King Saud University, P.O. Box 2460, Riyadh 11451, Saudi Arabia; mseleiman@ksu.edu.sa (M.F.S.); malotaibia@ksu.edu.sa (M.A.); ialashkar@ksu.edu.sa (I.A.-A.); 43910662@ksu.edu.sa (M.M.); 4Department of Crop Sciences, Faculty of Agriculture, Menoufia University, Shibin El-kom 32514, Egypt; 5School of Agriculture, Yunnan University, Kunming 650504, China; muzammal@ynu.edu.cn; 6Department of Botany, Government College University Allama Iqbal Road, Faisalabad 38000, Pakistan; nudrataauaf@yahoo.com; 7Agronomy Department, Faculty of Agriculture, Al-Azhar University, Cairo 11651, Egypt

**Keywords:** jute varieties, copper stress, phytoremediation, bioaccumulation factor, translocation factor, oxidative stress, growth

## Abstract

The current study was conducted to explore the potential for phytoremediation in different varieties of jute grown under toxic concentrations of copper (Cu). For this purpose, a Petri dish experiment was conducted under controlled conditions using four varieties of jute, i.e., HongTieGuXuan, C-3, GuBaChangaJia, and ShangHuoMa, grown in double filter paper under 50 µmol L^−1^ of artificially spiked copper (Cu) using CuSO_4_.H_2_O. The results of the present study revealed that jute varieties C-3 and HongTieGuXuan were able to survive under high concentrations of Cu without a significant decrease in plant height, plant fresh and dry weights, total chlorophyll content, or seed germination, while varieties GuBaChangaJia and ShangHuoMa exhibited a significant reduction in their growth and biomass. Furthermore, high concentrations of Cu in the medium resulted in lipid peroxidation. This could be due to the oxidative damage induced in the roots and leaves of the jute varieties, which might be a result of by hydrogen peroxide (H_2_O_2_) and electrolyte leakage. Reactive oxygen species (ROS) generated due to Cu toxicity can be overcome by the increasing activity of antioxidants, and it was also noted that GuBaChangaJia and ShangHuoMa exhibited high Cu stress, while C-3 and HongTieGuXuan showed some resistance to Cu toxicity. Contrastingly, Cu accumulation and uptake was higher in C-3 and HongTieGuXuan, while a little Cu was accumulated in the roots and leaves of GuBaChangaJia and ShangHuoMa. On the basis of these findings, it can be suggested that C-3 and HongTieGuXuan have the potential to cope with Cu stress and can be considered Cu-resistant varieties, while GuBaChangaJia and ShangHuoMa are considered Cu-sensitive varieties. Moreover, C-3 and HongTieGuXuan have the potential to revoke large amounts of Cu, and can be cultivated as phytoremediation tools in Cu-contaminated soil.

## 1. Introduction

“Heavy metals” is a general term which applies to metals and metalloids with atomic densities greater than 4000 kg m^−3^. Almost all the heavy metals are toxic to human beings even at low metal ion concentrations [1,2,3]. Heavy metals such as copper (Cu), mercury (Hg), lead (Pb), cadmium (Cd), and zinc (Zn) are harmful to the living organisms, including plants, when their concentrations increase beyond the permissible limits. Copper (Cu) is an important micronutrient present in many minerals and rocks, and also required for many metabolic processes in all living organisms, i.e., prokaryotes and eukaryotes [3,4,5,6]. Furthermore, Cu functions in many different enzymes including hemocynins, oxidases, and reductases, and also takes part in many biological and physiological processes in plants [5,7,8,9,10]. Contrastingly, excess Cu in the soil causes adverse effects on plants such as alteration in DNA structure, affected photosynthesis and respiration, stunted growth, chlorosis, and poor root development in higher plants [11,12,13,14,15,16]. Furthermore, phytotoxicity of Cu causes injury at the cellular level to many cellular organelles, which causes the generation of reactive oxygen species (ROS) [4,6]. Moreover, high concentrations of Cu in the soil enhance the production of ROS by producing superoxide radicals (O^−1^) and singlet oxygen (^1^O_2_) in the Mehler reaction [5,8], which leads to lipid peroxidation and the production of high contents of malondialdehyde (MDA) in plant tissues/cells, which induces oxidative damage in plants due to the phytotoxicity of Cu [7,13,17,18]. ROS production in plants is removed by a variety of antioxidant enzymes such as SOD, POD, CAT, and APX. Plant responses to oxidative stress also depend upon plant species and cultivars [7,9,19,20]. Previously, increasing antioxidant activity has been observed in *Raphanus sativus* [20], *Vitis vinifera* [18], and *Brassica napus* [14] under toxic concentrations of Cu in the soil.

Phytoremediation is the use of green plants) to remove, contain, and render harmful or toxic pollutants from the soil or medium. Furthermore, it is cost-effective, eco-friendly or with few environmental side effects, and is a publicly accepted technique which can also apply on specific sites or over large areas [8,13,21,22]. At present, 25 different plant species have been used as Cu accumulators (able to accumulate large amounts of metal in their above-ground parts rather than in below-ground parts) and Cu excluders (able to accumulate large amounts of metal in their below-ground parts rather than in above-gound parts) have been identified [7,8]. Previously, many different fibrous species, e.g., *Boehmeria nivea*, *Linum usitatissimum*, and *Hibiscus cannabinus*, been used for the phytoextraction of different heavy metals (Cu, Cd, and Pb) [11,23,24]. Among different fibrous species, jute (*Corchorus capsularis* L.) has been found to be relatively tolerant to heavy metal stress, possibly due to its huge biomass and specific physiological and biochemical activities [25,26,27]. Jute is a C-3 plant belonging to the family Malvaceae, and is cultivated as a natural fiber and also known as the golden fiber throughout the world. It is the cheapest and strongest fiber among all the natural fibrous crops, and main exporters of its fibers are India, Bangladesh, China, Thailand, and many other countries of southwest Asia [28]. Raw jute has various uses in making twine, matting, bales of cotton, sacks, coarse clothes, ropes, chair coverings, carpets, area rugs, curtains, and backing for linoleum [25,28].

Keeping in view the importance of jute, the present experiment was planned to explore the effect of Cu toxicity on the germination, seedling growth, biomass, antioxidant capacity, and Cu uptake of different varieties of jute. Sufficient literature is available on the phytotoxicity of Cu in *Raphanus sativus*, *Brassica juncea*, and *Phyllostachys pubescens* [8,15,20], but very few studies have been conducted on jute to study its different morphological and physiological traits. Findings from the present study will add to our understanding as to which varieties of jute can tolerate the toxic effects of Cu when grown in Cu-contaminated soil. The results from this study will add to our knowledge about (i) phytoremediation potential and (ii) the morpho-physiological traits and biochemical response of different varieties of jute grown in Cu-contaminated soil.

## 2. Materials and Methods

### 2.1. Growth Conditions and Experimental Treatment 

The seeds of different jute varieties were collected from Bast and Fiber Research Center of Huazhong Agricultural University, Hubei Province, China. The seeds of C-3, HongTieGuXuan, GuBaChangaJia, and ShangHuoMa (which are types of white jute) were used for the Petri dish experiment, and 40 seeds were planted in each Petri dish. Two filter papers (90 mm in diameter) were used in each Petri dish and 5 mL of 50 µmol L^−1^ Cu solution was added. Cu solutions were prepared with pure distilled water using copper sulfate (CuSO_4_.5H_2_O) (99% purity). After quantification of Cu, as percent availability in CuSO_4_.5H_2_O, 50 µmol L^−1^ doses of this compound were taken. Cu solution was applied every alternate day for the prevention of fungal infection and other contamination [29]. After washing carefully with distilled water, seeds were tested on filter paper (What-man No. 1). Seeds were surface sterilized with 0.1% HgCl_2_ for the prevention of surface fungal/bacterial contamination [30]. The experiment was conducted in March 2018 at Huazhong Agricultural University, Wuhan China, in a growth chamber under white lights (100 W, Guangdong PHILIPS Co., Guangdong, China) with a day/night temperature of 25 ± 2 °C and day/night humidity of 80%. The nutrient solution was provided once in a week, replacing the Cu treatment for 24 h. The experiment was conducted in a complete randomized design with nine replications for each treatment. The seed germination was recorded at 4 days after sowing (DAS). The seeds were considered to have germinated when the shoots were more than 2 mm [10], and plant height, plant fresh and dry weight, chlorophyll content, and different antioxidants were analyzed at 14 DAS. Furthermore, the Cu concentration of roots and shoots were also measured in this study. All chemicals used were of analytical grade, procured from Sinopharm Chemical Reagent Co., Ltd (Shanghai, China).

### 2.2. Growth and Morphological Traits

Different morphological attributes were noted after the harvest of all plants. Plants in each treatment were harvested and separated into roots and shoots for growth and morphology traits. Total length was defined as the length of the plant from the surface growth medium line of the Petri dish to the tip of the uppermost shoot. Total fresh weight was measured by measuring the weight of roots and shoots with the help of a digital weighing balance. After that, plant samples were oven dried for 1 h at 105 °C, then 65 °C for 72 h until the weight become uniform and dry biomass was recorded. Roots were washed with distilled water and dipped in 20 mM Na_2_EDTA for 15–20 min, washed thrice with distilled water and finally with de-ionized water, and then oven dried for further analysis. Chlorophyll contents were determined following the method of Arnon [31] and are expressed as mg g^−1^ FW.

### 2.3. Determination of Oxidative Stress 

The method used to describe the concentration of lipid peroxidation was presented by Heath and Packer [32] and expressed as μmoles g^−1^ FW.

For the estimation of H_2_O_2_ content, an H_2_O_2_ Assay Kit (Suzhou Comin Biotechnology Co., Ltd.) was used.

The electrolyte leakage (EL) was measured according to the standard procedure of Dionisio-Sese and Tobita [33].

### 2.4. Analysis of Antioxidant Enzyme Activities

The activity of SOD was measured by the method of Chen and Pan [34] and expressed in Ug^−1^ FW.

The activity of POD was measured by the method of Sakharov and Aridilla [35] and expressed in Ug^−1^ FW.

The enzymatic activity of CAT was measured by the method of Aebi [36] and expressed in Ug^−1^ FW.

Ascorbate peroxidase activity was measured according to Nakano and Asada [37] and expressed in Ug^−1^ FW.

### 2.5. Determination of Cu Concentration

The dried samples were ground into powdered form using stainless steel and 0.1 g of root and shoot samples was taken for digestion in HNO_3_/HClO_4_ (4:1) solution. Final readings were taken from an atomic absorption spectrophotometer (AAS) model Agilent 240 FS-AA [7].

Bioaccumulation factor (BAF) was measured as the proportion of Cu concentration in plant tissues and Cu concentration in medium using the following formula: (1)$$\mathrm{BAF}=\frac{\mathrm{Cu}\;\mathrm{concentration}\;\mathrm{in}\;\mathrm{plant}\;\mathrm{tissues}}{\mathrm{Cu}\;\mathrm{concentration}\;\mathrm{in}\;\mathrm{the}\;\mathrm{medium}}$$

Translocation factor (TF) was evaluated as the proportion of Cu concentration in shoots with respect to the roots:(2)$$\mathrm{TF}=\frac{\mathrm{Cu}\;\mathrm{concentration}\;\mathrm{in}\;\mathrm{shoots}}{\mathrm{Cu}\;\mathrm{concentration}\;\mathrm{in}\;\mathrm{the}\;\mathrm{roots}}$$

### 2.6. Statistical Analysis

The data recorded were statistically analyzed using Statistix 8.1 (Analytical Software, Tallahassee, FL, USA). Testing showed that all the data were approximately normally distributed. Thus, the differences between treatments were determined using analysis of variance, and the least significant difference test (*p* ≤ 0.05) was used for multiple comparisons between treatment means. Graphical representation was carried out using SigmaPlot 12 and R studio. 

## 3. Results 

### 3.1. Effects of Cu Toxicity on Seed Germination, Growth, and Chlorophyll Content

Jute varieties showed various germination behaviors in response to Cu exposure (Table 1). Germination rate ranging from 75–100% in different varieties of jute under Cu toxicity. Based on the observations of seed germination, HongTieGuXuan and C-3 showed a better germination rate than GuBaChangaJia and ShangHuoMa. Maximum germination percentage was observed in HongTieGuXuan (100%) and C-3 (100%), followed by GuBaChangaJia (77.5%) and ShangHuoMa (75%) under toxic concentrations of Cu in the medium. 

In the present study, growth in terms of plant height, plant fresh weight, and plant dry weight was also measured in different varieties of jute when cultivated under high concentrations of Cu in the medium. Results regarding growth and biomass of different varieties of jute are presented in Table 1. Maximum plant height and fresh and dry biomass were observed in HongTieGuXuan and C-3 under toxic levels of Cu in the medium. Moreover, the minimum plant height (0.61 ± 0.13), plant fresh weight (0.101 ± 0.003), and plant dry weight (0.040 ± 0.02) was observed in ShangHuoMa, while maximum plant height (3.73 ± 0.85), plant fresh weight (0.30 ± 0.01), and plant dry weight (0.100 ± 0.01) were observed in C-3 under toxic concentrations of Cu in the medium.

In the present study, total chlorophyll contents of the leaves of different varieties of jute are presented in Table 1. Different chlorophyll contents were observed in different varieties to jute after exposure of high concentrations of Cu in the medium. Maximum contents of chlorophyll were observed in HongTieGuXuan and C-3. The maximum contents of chlorophyll were observed in C-3 (2.82 ± 0.03 mg g^−1^ FW) followed by HongTieGuXuan (2.64 ± 0.09 mg g^−1^ FW), while minimum contents of chlorophyll were observed in ShangHuoMa (1.47 ± 0.08 mg g^−1^ FW) followed by GuBaChangaJia (1.51 ± 0.03 mg g^−1^ FW). 

### 3.2. Effect of Cu Toxicity on Oxidative Stress

In this study, the effects of Cu toxicity on different varieties of jute on malondialdehyde (MDA), hydrogen peroxide (H_2_O_2_), and electrolyte leakage (EL) from the roots and shoots were also investigated (Figure 1). Exposure of Cu concentration to different varieties of jute significantly increased the MDA, H_2_O_2_, and EL in the roots and leaves of different varieties of jute (Figure 1). The contents of MDA, H_2_O_2_, and EL were higher in the roots when compared to the above-ground parts of the plant. According to the results, maximum contents of MDA (17.7 ± 0.8 μmoles g^−1^), H_2_O_2_ (539 ± 6 μmoles g^−1^), and EL (70 ± 2%) were observed in the roots of ShangHuoMa. Similarly, in the shoots, the maximum contents of MDA (11.9 ± 0.2 μmoles g^−1^), H_2_O_2_ (445 ± 5 μmoles g^−1^), and EL (56 ± 2%) were also observed in of ShangHuoMa compared to other varieties of jute.

### 3.3. Effect of Cu Toxicity on Antioxidant Activities

In the present study, the antioxidants superoxidase dismutase (SOD), peroxidase (POD), catalase (CAT), and ascorbate peroxidase (APX) were also investigated in different varieties of jute under toxic concentrations of Cu in the medium (Figure 2). Activities of antioxidant were higher in GuBaChangaJia and ShangHuoMa than in C-3 and HongTieGuXuan. The maximum activity of SOD (90 ± 1.4 U g^−1^ FW), POD (3400 ± 138 U g^−1^ FW), CAT (222 ± 5 U g^−1^ FW), and APX (441 ± 4 U g^−1^ FW) was observed in the roots of ShangHuoMa. Similarly, in the shoots, the maximum activity of SOD (86 ± 1.4 U g^−1^ FW), POD (2920 ± 23 U g^−1^ FW), CAT (191 ± 3 U g^−1^ FW), and APX (375 ± 4 U g^−1^ FW) were also observed in ShangHuoMa.

### 3.4. Cu Uptake and Bioaccumulation in the Roots and Shoots

In this study, the concentrations of Cu from different parts (roots and shoots) of jute varieties were also determined under toxic concentrations of Cu in the medium. The concentrations of Cu in the roots and shoots of different varieties of jute are presented in Table 2. These results suggested that Cu concentration was higher in the varieties that exhibited significantly better growth than the varieties that were more affected by Cu stress. The concentration of Cu in the roots ranged from 37 to 60 mg kg^−1^, while in the shoots, Cu concentration ranged from 38 to 61 mg kg^−1^ (Table 2). The maximum Cu was accumulated in the roots of C-3 (60 ± 0.8 mg kg^−1^), followed by HongTieGuXuan (56 ± 1.4 mg kg^−1^) and GuBaChangaJia (40 ± 0.8 mg kg^−1^). Similarly, in the shoots, maximum Cu concentration was accumulated in C-3 (61 ± 0.9 mg kg^−1^), followed by HongTieGuXuan (57 ± 1.1 mg kg^−1^) and GuBaChangaJia (41 ± 0.6 mg kg^−1^).

In this study, bioaccumulation factor (BAF) and translocation factor (TF) were also calculated (Figure 3). It was noticed that the values of BAF and TF of HongTieGuXuan and C-3 were greater than 1 while the values of BAF and TF of GuBaChangaJia and HongTieGuXuan were less than 1 (Figure 3). The maximum BAF value was shown by C-3 in the roots (1.19) and shoots (1.20), compared to other varieties of jute. Contrastingly, the minimum BAF value was shown by HongTieGuXuan in the roots (0.74) and shoots (0.74), compared to other varieties of jute. Similarly, the maximum value of TF was also shown by C-3 (1.01), followed by HongTieGuXuan (1.01) and GuBaChangaJia (0.99).

### 3.5. Correlation between Growth, Biomass, Total Chlorophyll Content, and Cu Uptake

A Pearson’s correlation analysis was carried out to quantify the relationship between growth, biomass, total chlorophyll content, and Cu uptake in the roots and shoots of different varieties of jute (Figure 4). Cu concentration in the roots was positively correlated with Cu concentration in the shoots, and also positively correlated with growth parameters and chlorophyll content. In the same way, Cu concentration in the shoots was also strongly positively correlated with plant height, fresh and dry biomass, and chlorophyll content. This correlation reflected the close connection between Cu uptake and growth in different varieties of jute.

## 4. Discussions

In the last few decades, soil concentrations of Cu have surpassed toxic levels (>30 mg kg^−1^) due to overpopulation and industrialization [5,6,15,38]. It has been observed that a reduction in plant growth and biomass in a common response of plants to Cu stress [4,8,39]. For the successful production of jute when cultivated in the Cu-contaminated soil of China, development and selection of tolerant varieties through screening is crucial. For better growth and development of jute for fiber extraction and natural resources, it is necessary to cultivate the most suitable variety of jute [25,26]. However, the resistance or tolerance mechanism of a plant depend upon its specific physiological and biochemical activities [14,40,41]. Therefore, a preliminary experiment was conducted to expose Cu-sensitive varieties and Cu-resistant varieties to toxic concentrations of Cu in the medium.

The germination assay is a basic method to observe the effects of heavy metal stress on plant seedlings. Moreover, seed germination is one of the most important biological parameters in the life cycle of a plant [10,16,42,43]. Seed germination of different varieties of jute exhibited different responses to exposure to Cu stress via the medium (Table 1). It was also noticed that HongTieGuXuan and C-3 showed the maximum germination percentage, while GuBaChangaJia and ShangHuoMa showed the minimum germination percentage, which might be due to high toxicity caused by the Cu in the medium. Previously, it has also been suggested that germination percentage at seedling stage can be affected by the phytotoxicity of Cu, which might be due to the accumulation of carbon partitioning in the tissues of the plant [10,16]. Furthermore, the minimum germination rate in GuBaChangaJia and ShangHuoMa might be due to water deficiency due to excess Cu in the medium inhibiting cell expansion and reducing carbon assimilation [10,44]. These results coincided with Nizam et al. [42], who found that high concentrations of As reduced the germination percentage of some varieties, while others showed a better germination rate compared to the control.

High concentrations of Cu are extremely toxic for growth and biomass of plants. It was noticed that dominant plants collected from Cu mining sites showed more resistance than normal plants [8,39]. In the present study, HongTieGuXuan and C-3 showed better growth in terms of plant height and plant fresh and dry biomass compared to GuBaChangaJia and ShangHuoMa (Table 1). The poor growth and composition in GuBaChangaJia and ShangHuoMa under high concentrations of Cu in the medium can be attributed to insufficient uptake of nutrients, low availability of water, perturbed root architecture, and poor stomata regulation of plant metabolic processes [45]. Furthermore, nutrient acquisition, stimulation of the defense system (antioxidants), structural integrity of metabolites, and considerable water use efficiency are positively associated with heavy metal stress tolerance in different plant species [10,46,47]. Similar findings were shown by Uddin et al. [48] when they studied different varieties of jute and noticed that different varieties exhibited different responses to exposure to high concentrations of Pb in their soil. 

Copper is a micronutrient that provides support to the shaping and function of chloroplasts in plant cells [4], but excess Cu affects chloroplast structure and ultimately reduces the photosynthetic pigment of the plants [49,50]. In the present study, excess Cu in the medium caused a drastic reduction in chlorophyll content in Cu-sensitive varieties (Table 1). The reduction of chlorophyll content in GuBaChangaJia and ShangHuoMa under toxic concentration of Cu in the medium might be due to the inhibited activities of various enzymes associated with chlorophyll biosynthesis [11,50]. Furthermore, the accumulation of Cu concentrations in the tissues of GuBaChangaJia and ShangHuoMa also caused low chlorophyll content in the leaves of these jute varieties.

Heavy metal stress can alter the equilibrium of reactive oxygen species (ROS) production, which promotes membrane lipid peroxidation and ROS accumulation, and disturbs the function and structure of the cell membrane [20,23,39]. ROS have also been shown to play a role in ABA mediated root growth in Arabidopsis [50,51]. Moreover, the generation of ROS under high concentrations of Cu in the soil is enhanced by cuprous and cupric Cu ions, which induce oxidative damage in plant cells/tissues [9,18,52]. MDA is an oxidized product of membrane lipids and is supported by leakage of plasma membrane and cell turgor loss [39,53]. The production of ROS in plant cells/tissues is very dangerous, and plants have evolved special defense systems such as SOD, POD, CAT, and APX to scavenge ROS [7,8,14]. Plant responses to oxidative stress depend on the plant species and variety [47,54]. It was noticed that Cu-sensitive (GuBaChangaJia and ShangHuoMa) varieties underwent high oxidative stress, as shown by high contents of MDA, H_2_O_2_, and EL (Figure 1). Similar results were suggested by Khan et al. [47]: that tolerant varieties undergo less oxidative stress when compared to sensitive varieties. Furthermore, it was also observed that to overcome oxidative stress, plants have a special defense system to scavenge ROS generation (Figure 2). GuBaChangaJia and ShangHuoMa showed more antioxidant activity than HongTieGuXuan and C-3, which might be due to more oxidative stress in these species. The difference in antioxidant activities might be due to species-specific biochemical responses, as shown by Akram et al. [45]. 

Metal accumulation in different parts of a plant depends upon plant species, growth stage, fertilizer application, and growth conditions [4,7]. Based on tolerance mechanisms, plant species can been divided into two types: (1) Metal excluders accumulate heavy metals from the substrate in their roots, but restrict their transport and entry into their aerial parts; (2). Hyperaccumulators are able to accumulate large amounts of metals in their above-ground parts rather than in belowground parts [55]. Furthermore, bioaccumulation factor (BAF) and translocation factor (TF) are important in screening hyperaccumulators for phytoremediation of heavy metals. Screening of hyperaccumulators depends upon BAF and TF (both should be greater than 1) for evaluation and selection of plants for phytoremediation [8,25,56,57,58]. In many previous studies, jute has been considered as a hyperaccumulator species for different heavy metals; for example, Ahmed and Salima [25] studied jute under different heavy metal exposures and noticed that is a hyperaccumulator for different heavy metals including Cd, Cu, Cr, and Pb. Our results suggested that HongTieGuXuan and C-3 are able to accumulate large amounts of Cu in their above-ground parts and can be considered hyperaccumulator varieties, while GuBaChangaJia and ShangHuoMa are able to accumulate low Cu concentrations in their above-ground parts and can be considered Cu-excluder species. Furthermore, the values of BAF and TF of HongTieGuXuan and C-3 were greater than 1, while the values of BAF and TF for GuBaChangaJia and ShangHuoMa were less than 1. Phytoremediation potential of different varieties of jute has already been studied by Uddin et al. [48] under lead-contaminated soil, and it was found the similar results that Pb-tolerant varieties were good hyperaccumulator species (showed values of BAF and TF greater than 1) and were able to accumulate large amount of Pb in their above-ground parts rather than in the below-ground parts of the plant. 

## 5. Conclusions

Overall, Cu stress significantly decreased growth and development (seed germination, plant height, fresh and dry biomass, chlorophyll content) in all varieties of jute, but Cu tolerance index was higher in HongTieGuXuan and C-3 than in GuBaChangaJia and ShangHuoMa. However, jute has an active antioxidative defense system to scavenge the ROS produced due to high concentrations of Cu in the medium. The results suggested that HongTieGuXuan and C-3 are Cu-tolerant varieties, while GuBaChangaJia and ShangHuoMa are Cu-sensitive varieties. Furthermore, the accumulation of Cu in different parts (roots and shoots) of the plants also indicated that HongTieGuXuan and C-3 can be considered Cu hyperaccumulator species, while GuBaChangaJia and ShangHuoMa can be considered Cu excluder species. However, future research is needed to study the effects of Cu stress on different varieties of jute for fiber extract in field areas. 

## Figures and Tables

**Figure 1 plants-08-00545-f001:**
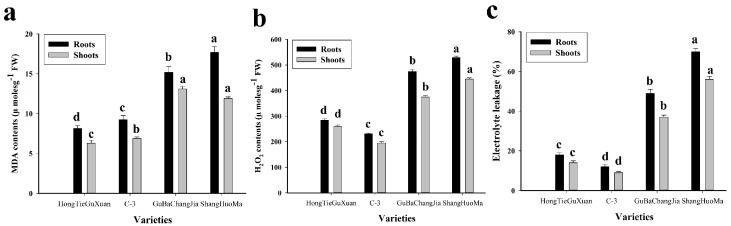
Effect of Cu stress on malondialdehyde (MDA) (**a**), H_2_O_2_ (**b**) and electrolyte leakage (EL) (**c**) in different varieties of jute. Values in the figures represent just one harvest. Different letters within a column indicate significant differences between the treatments (*p* < 0.05). Bars indicate the mean ± SD (*n* = 3).

**Figure 2 plants-08-00545-f002:**
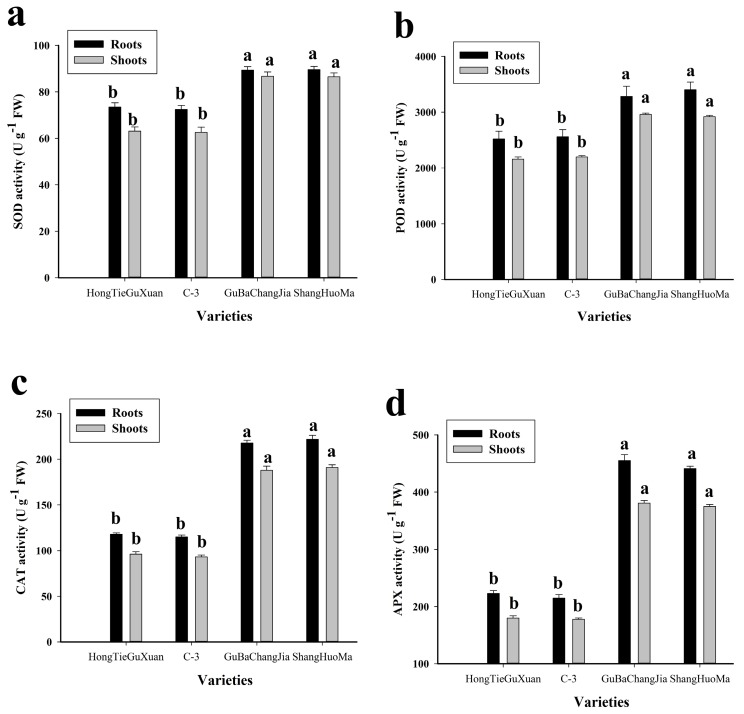
Effect of Cu stress on superoxidase dismutase (SOD) (**a**), peroxidase (POD) (**b**), catalase (CAT) (**c**), and ascorbate peroxidase (APX) (**d**) in different varieties of jute. Values in the figures represent just one harvest. Different letters within a column indicate significant differences between the treatments (*p* < 0.05). Bars indicate the mean ± SD (*n* = 3).

**Figure 3 plants-08-00545-f003:**
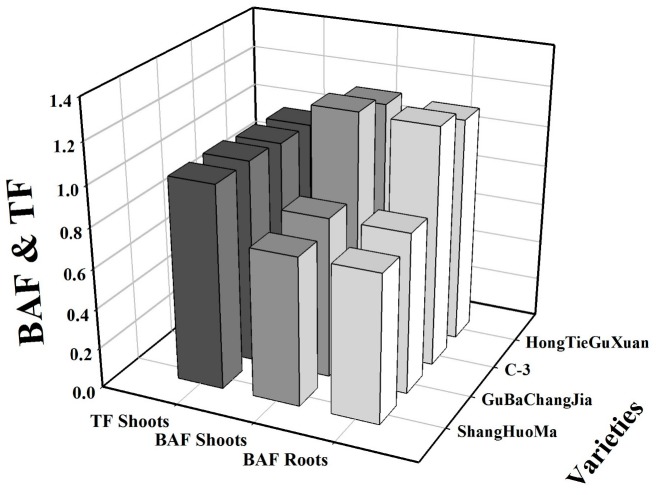
Effect of Cu stress on bioaccumulation factor (BAF) and translocation factor (TF) in different varieties of jute.

**Figure 4 plants-08-00545-f004:**
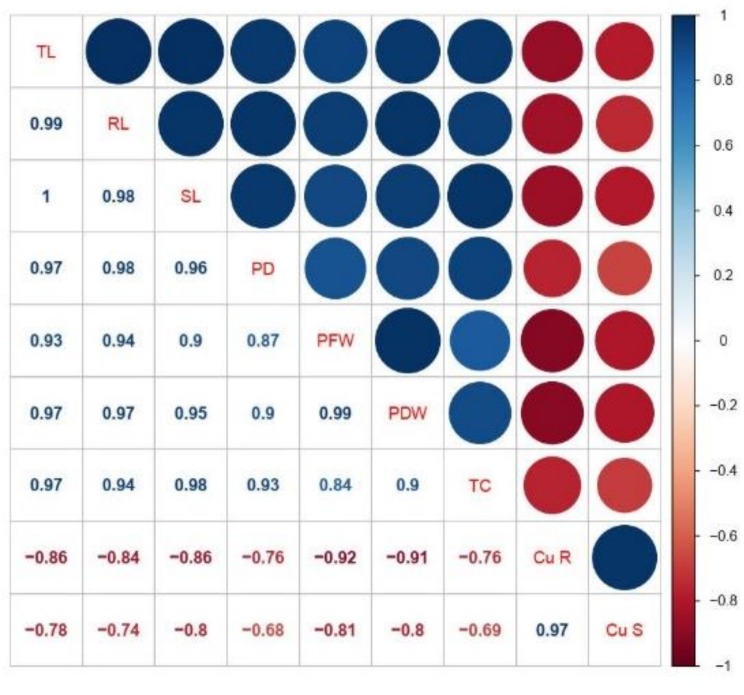
Correlation of Cu uptake, with growth parameters and total chlorophyll content. Cu R: Cu concentration in the roots; Cu S: Cu concentration in the shoots; SL: shoot length; RL: root length; TL: total length; PD: plant diameter; PFW: plant fresh weight; PDW: plant dry weight; and TC: total chlorophyll content.

**Table 1 plants-08-00545-t001:** Effect of Cu stress on plant height (cm), plant fresh weight (g), plant dry weight (g), total chlorophyll content (mg g^−1^ FW), and seed germination (%) in different varieties of jute.

Varieties	Plant Height	Plant Fresh Weight	Plant Dry Weight	Total Chlorophyll	Seed Germination
HongTieGuXuan	3.72 ± 0.05 a	0.255 ± 0.05 b	0.103 ± 0.005 a	2.64 ± 0.09 a	100
C-3	3.73 ± 0.85 a	0.3 ± 0.01 a	0.100 ± 0.01 a	2.82 ± 0.03 a	100
GuBaChangaJia	0.87 ± 0.81 b	0.103 ± 0.003 c	0.045 ± 0.015 b	1.51 ± 0.03 b	77.5
ShangHuoMa	0.61 ± 0.13 b	0.105 ± 0.003 c	0.040 ± 0.02 b	1.47 ± 0.08 b	75

Values in the table are from one harvest ± SD (*n* = 5). Different letters within a column indicate significant difference between the treatments (*p* < 0.05).

**Table 2 plants-08-00545-t002:** Accumulation of Cu in roots (mg kg^−1^) and shoots (mg kg^−1^) of different varieties of jute.

Varieties	Cu Concentration in Roots	Cu Concentration in Shoots
HongTieGuXuan	56 ± 1.4 b	57 ± 1.1 b
C-3	60 ± 0.8 a	61 ± 0.9 a
GuBaChangaJia	40 ± 0.8 c	41 ± 0.6 c
ShangHuoMa	37 ± 1 d	38 ± 0.9 d

Values in the table represent just one harvest (mean ± SD) (*n* =3). Different letters within a column indicate significant differences between the treatments (*p* < 0.05). Relative radiance of plastic filter used: HongTieGuXuan, C-3, GuBaChangaJia, and ShangHuoMa.

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
