# Peer review of "Assessing the Correlations between Different Traits in Copper-Sensitive and Copper-Resistant Varieties of Jute (Corchorus capsularis L.)"

_plants, 2019, doi:10.3390/plants8120545_

Round 1
Reviewer 1 Report
This manuscript presents investigation of morphological changes, antioxidant response and phytoextraction of copper in different varieties of jute seedlings. Overall, the manuscript is an interesting contribution to scientific knowledge in the field of studies of plant resistance to heavy metals pollution and development of phytoremediation technologies.
Nevertheless, several problems/doubts should be solved before the manuscript is suitable to be published.
Most serious problems/doubts are caused by the design of the experiment, in particular the missing of the control. Perhaps the observed differences in values (plant fresh weight, seed germination, chlorophyll concentration, enzymes activities) between varieties are associated not with the toxicity of copper, but with features in the growth of plants of these varieties (GuBaChangaJA and ShangHuoMa)? The missing of control plants casts doubt on the obtained results and the discussion of the results given by the authors. The authors have studied four varieties of jute, but they did not mention anywhere what species of plants they worked with. There are different species of jute, so it must be added to manuscript. If studied varieties belong to different species, the differences in the reaction of plants to Cu pollution could be related to species specificity and should be discussed additionally. Please provide how many days in total the plants were grown on medium containing copper. It is not clear from Statistical analysis description, why for analysis of variance the authors used Kolmogorov-Smirnov test (nonparametric test), but for correlation analysis the authors used Pearson correlation (parametric test)? How many replications were conducted in this study? The manuscript contains conflicting information. For example, L105: “…with nine replications for each treatment”, L 148: “All data presented in this study are the means of three replicates”, L 163: “Values in the table is just one harvest (mean ± SD) (n =5)”. Please check these, revise or explain what you mean under replicates in each case. Please specify how to calculate the BAF. In formula in manuscript as denominator is written “Cu concentration in the soil”. Bur the authors did not work with soil. What value for the denominator did the authors use? Have μmoles been converted to mg? The concentrations of copper in shoots and roots are presented in mg per kg. L344: Conclusion section. The authors mentioned about “Cu tolerance index”. Please provide the formula for calculation of this index.Minor concerns
The first sentence in Introduction section must be rephrased due to some reasons. Firstly, phrase “heavy metal are the group of heavy metals” is unclear and has a grammar error. Secondly, “almost all the heavy metal are toxic for the plants and animals, even in low concentration” is contrary to subsequent information on the benefits of copper in low concentrations for living organisms. L 53: what means “choloration”? L60-L61: “….superoxide dismutase (SOD), peroxidase (POD), catalase (CAT) and ascorbate peroxidase (APX) plays many important roles in reducing the metal toxicity in the plants by secreting some chemicals”. English is not good! It is not clear which chemicals are secreting by these enzymes. L62-63: “Previously, it was reported that the activities of antioxidants plant an important role…”. English is not good! L71: “...nivea. Linum usitatissimum and…”. Please replace full stop to comma. L77: “…and many other countries of southwest Asia including Brazil”. Brazil does not belong to Asia. Please correct this sentence. In the notes to the tables and in the legends to the figures the authors used the phrase «Relative radiance of plastic filter used: HongTieGuXuan, C-3, GuBaChangaJA and ShangHuoMa”. It is not clear what the authors mean under “Relative radiance of plastic filter used”?Author Response
This manuscript presents investigation of morphological changes, antioxidant response and phytoextraction of copper in different varieties of jute seedlings. Overall, the manuscript is an interesting contribution to scientific knowledge in the field of studies of plant resistance to heavy metals pollution and development of phytoremediation technologies.
Nevertheless, several problems/doubts should be solved before the manuscript is suitable to be published.
Major comments:
Most serious problems/doubts are caused by the design of the experiment, in particular the missing of the control. Perhaps the observed differences in values (plant fresh weight, seed germination, chlorophyll concentration, enzymes activities) between varieties are associated not with the toxicity of copper, but with features in the growth of plants of these varieties (GuBaChangaJA and ShangHuoMa)? The missing of control plants casts doubt on the obtained results and the discussion of the results given by the authors.
Response: Respected Reviewer, as it is just a correlation between four different varieties of jute without any control. In this study, our objective is that which variety can tolerate better under Cu stress condition and which variety can be used as phytoremediation material for Cu in cu stress environment. Although we are determining the differences between growth between these four varieties and correlate with each other and conclude our results on the behalf of their correlation. In the results section we write it very clearly that this variety is increased from that one. It’s simply means that we are correlating four different varieties to conclude that from these four varieties these can grow better in Cu contaminated area. Furthermore, this kind of study was not conducted from time, many literature are available in which Authors are just correlating their varieties/genotypes with each other without use of any control. Pak. J. Bot., 45(4): 1241-1245, 2013. 2017. eISSN 2249 0256. Romanian Biotechnological Letters 23(2) · https://doi.org/10.1016/j.plaphy.2018.04.001. Pak. J. Bot., 49(6): 2201-2212,
Furthermore, for better understanding I also change the title of the manuscript.
Major comments:
The authors have studied four varieties of jute, but they did not mention anywhere what species of plants they worked with. There are different species of jute, so it must be added to manuscript. If studied varieties belong to different species, the differences in the reaction of plants to Cu pollution could be related to species specificity and should be discussed additionally.
Response: Respected Reviewer, in this study we used four varieties of jute i.e. HongTieGuXuan, C-3, GuBaChangaJA and ShangHuoMa. There are two different types 1. Corchorus capsularis known as white jute (a variety originates from poor villagers of India), and 2. Corchorus olitorius known as Tosaa jute (a variety originates through native South Asia). For the best of our knowledge, these four varieties belongs to Corchorus capsularis white jute and we write in the manuscript.
Major comments:
Please provide how many days in total the plants were grown on medium containing copper.
Response: Respected Reviewer, in the L 108, we have write about this detail that plants were grow for 14 days in the petri dishes.
Major comments:
It is not clear from Statistical analysis description, why for analysis of variance the authors used Kolmogorov-Smirnov test (nonparametric test), but for correlation analysis the authors used Pearson correlation (parametric test)?
Response: Respected Reviewer, I rewrite it and removed this test “Kolmogorov-Smirnov test (nonparametric test)” and write in easy way and always mention about R Studio.
Major comments:
How many replications were conducted in this study? The manuscript contains conflicting information. For example, L105: “…with nine replications for each treatment”, L 148: “All data presented in this study are the means of three replicates”, L 163: “Values in the table is just one harvest (mean ± SD) (n =5)”. Please check these, revise or explain what you mean under replicates in each case.
Response: Respected Reviewer, We have used 40 seeds in each petri dish with nine replication as mention in L105. L 148 it was our fault now its corrected while L 163, Sir its means that we selected 5 randomly plants for morphological traits from different replications and in the figures we used (n=3) it means we used three replications. So according to your comments it’s now corrected,
Major comments:
Please specify how to calculate the BAF. In formula in manuscript as denominator is written “Cu concentration in the soil”. Bur the authors did not work with soil.
Response: Respected Reviewer, Bioaccumulation factor (BAF) was measured as the proportion of Cu concentration in plant tissues and Cu concentration in medium by using the following formula: We have changed soil to medium. Thanks for correction
Major comments:
What value for the denominator did the authors use? Have μmoles been converted to mg? The concentrations of copper in shoots and roots are presented in mg per kg.
Response: Respected Reviewer, concentration of Cu is always in ppm, so we can use mg kg-1 as we used in the manuscript but μmoles we used for MDA contents not for Cu concentration. Although, throughout the manuscript we have checked it.
Major comments:
L344: Conclusion section. The authors mentioned about “Cu tolerance index”. Please provide the formula for calculation of this index.
Response: Respected Reviewer, although for the best of our knowledge there is no formula regarding Cu tolerance index while this kind of terminology has been used in many previous papers such as doi:10.1016/S2095-3119(13)60279-8 for determining the tolerance in two varieties.
Minor concerns
The first sentence in Introduction section must be rephrased due to some reasons. Firstly, phrase “heavy metal are the group of heavy metals” is unclear and has a grammar error. Secondly, “almost all the heavy metal are toxic for the plants and animals, even in low concentration” is contrary to subsequent information on the benefits of copper in low concentrations for living organisms.
Response: Respected Sir, I have rewrite this whole sentence now.
L 53: what means “choloration”?
Response: Respected Sir, it’s written by mistake. Actually it was chlorosis which means is a condition in which leaves produce insufficient chlorophyll. As chlorophyll is responsible for the green color of leaves, chlorotic leaves are pale, yellow, or yellow-white.
Major comments:
L60-L61: “….superoxide dismutase (SOD), peroxidase (POD), catalase (CAT) and ascorbate peroxidase (APX) plays many important roles in reducing the metal toxicity in the plants by secreting some chemicals”. English is not good! It is not clear which chemicals are secreting by these enzymes.
Response: Respected Sir, actually this sentences was incorrect and I rewrite it and marked red you can see now.
L62-63: “Previously, it was reported that the activities of antioxidants plant an important role…”. English is not good!
Response: Respected Sir, I have rewrite this sentence
L71: “...nivea. Linum usitatissimum and…”. Please replace full stop to comma.
Response: Done
L77: “…and many other countries of southwest Asia including Brazil”. Brazil does not belong to Asia. Please correct this sentence.
Response: Respected Sir, I have removed Brazil from this sentence
In the notes to the tables and in the legends to the figures the authors used the phrase «Relative radiance of plastic filter used: HongTieGuXuan, C-3, GuBaChangaJA and ShangHuoMa”. It is not clear what the authors mean under “Relative radiance of plastic filter used”?
Response: Respected Sir, “Relative radiance of plastic filter” we read this sentence from DOI 10.1007/s00344-016-9625-y this paper and they used as the abbreviation of the treatments. However, I have rewrite according to your comments and marked as red.
Reviewer 2 Report
Please find enclosed in the document.

Author Response
Overall a helpful and well written article due to its applied approach ‐ finding resistant varieties to copper. Minor changes should be done before publishing, regarding English language phrasing, grammar, typos, like: Line 25: “..can tolerate Cu stress” Line 30: …by high contents of hydrogen… Line 32: ..overcomes the activity of antioxidants..?! + delete “some” Line 35: ..have considerable.. Line 44: consider rephrase it; for instance, “almost heavy metals are toxic..concentration” Line 70: ….fibrous crops (species) have been used… Line 72: …different fibrous crop, jute has been found relatively. Line 76: ..cheapest and produces strongest fiber among.. Line 84: ..very few studies have explored Cu toxicity on… Line 93: …were used for…and 40 seeds were sowed in a plate (90 mm diameter) with filter paper and… Line 95: Please delete “in laboratory” Line 99: Delete double space before “seeds” Line 101: ..in March 2018.. Line 101‐102: (Philips…) under T = T 1 – T2 and HR = 80% dark/night, white light intensity xx. Please delete the rest and add the proper temperatures for day and night Line 112: ..after harvesting all plants. Delete the rest of the sentence. Line 116: weighing Line 138: …dried samples were grounded into powder and 0.1 g was digested in,…solution. Line 167: instead of “grown”, please write “cultivated” (throughout the text) Line 195: ..values in figures are obtained from one harvest only. Line 260: ..decades, concentrations of copper in…have reached.. Line 270: germination assay is a basic method.. Line 280: these results coincide with…who found that high concentrations of As reduce germination percentage… Line 249 ‐ 250: Uddin et al found that different varieties responded differently to exposure of.. Cut the second phrase, is redundant unless you give some data (values) Line 258 ‐ 259: Copper is a micronutrient providing support to.. Line 315: undergoes – undergo. Delete double space before “similar results”
Response: Respected Sir, thanks for your precious time. I have done all the change according to your comments and your comments are always important for me and I have done it throughout my manuscript.
Round 2
Reviewer 1 Report
Thanks to the authors for carefully correction of the manuscruipt according to the comments and for authors' responces. I believe that the manuscript has been significantly improved and now can be published in Plants.
Author Response
Thanks dear reviewer for recommending our MS for publication.